# Degree of Knowledge and Commitment of the Spanish Podiatry Community to Green Podiatry: A Preliminary Report

**DOI:** 10.3390/ijerph20186761

**Published:** 2023-09-14

**Authors:** Lara Gómez-Ruiz, Alfonso Martínez-Nova, Eduardo Simón-Pérez, Juan Francisco Morán-Cortés

**Affiliations:** 1Nursing Department, Universidad de Extremadura, Avda, Virgen del Puerto, 2, 10600 Plasencia Cáceres, Spain; 2Podiatry Grupo Recoletas, C/Angustias n 17, 47003 Valladolid, Spain

**Keywords:** climatic change, podiatry, green, shoes

## Abstract

Climate change is real and we are witnessing its consequences, such as rising temperatures, water scarcity, and sea-level rise, among other significant impacts. As healthcare professionals, podiatrists should be concerned about climate change; however, they still contribute to the damage caused. Therefore, the aim of this study was to assess the level of awareness among podiatrists regarding this issue and determine their attitudes toward the climate change process. The study involved conducting a survey comprising a series of questions, including personal information, Likert-scale questions, and short questions to evaluate attitudes toward environmental sustainability in their workplace and how they contribute to the climate change process. The questions addressed their commuting habits to determine sustainability, the number of hours of physical exercise per week, and their clinical attitudes, such as prescribing unnecessary treatments or emphasizing sports as the primary treatment. The results revealed that nearly 89% of the respondents were unaware of ecological podiatry or shoe recycling. Regarding clinical attitudes, 31.1% of the respondents stated that they prescribe sports as the primary treatment for all their patients, while 37.9% do so in most cases. They also tend to avoid prescribing unnecessary treatments, with 44.9% stating that they never prescribe them unless absolutely necessary. In conclusion, based on this survey, Spanish podiatrists do not undertake favorable actions for climate change and lack knowledge of the concept of “ecological podiatry”. To improve the situation, efforts should focus on waste reduction, eliminating unnecessary treatments, and promoting and practicing ecological podiatry.

## 1. Introduction

Healthcare is designed to promote good health, but it can clearly contribute to climate change, which is considered the greatest global health threat of the 21st century. Climate change impacts health deficits from multiple angles, primarily through the use of fossil fuels, which increases atmospheric carbon dioxide (as well as nitrous oxide and methane). These greenhouse gases trap heat, causing temperatures and sea levels to rise, leading to severe weather patterns including floods, storms, and droughts [1].

Those already at risk of health issues, such as the elderly, pregnant women, children, and those with broader health comorbidities, are particularly vulnerable. As a result, since 1995, a series of annual meetings (COP) have been conducted within the framework of the United Nations Framework Convention on Climate Change (UNFCCC), during which member countries discuss and undertake actions regarding global climate change. Topics include reducing greenhouse gas emissions, adapting to climate change effects, and enhancing the resilience of developing countries to improve their capacity to adapt to climate change impacts through international cooperation. This global recognition culminated in the Paris Agreement (COP21), the United Nations Sustainable Development Goals, and the World Health Organization’s Climate Change Action Plan. It was agreed that the majority of health impacts stem from energy and resource consumption, as well as greenhouse gas production in the environment. Numerous professional medical associations and healthcare providers advocate for their members to take the lead in environmental sustainability efforts [2]. In 2023, the first “world balance” will assess progress towards the goals of the Paris Agreement.

In 2020, the COVID-19 pandemic presented an opportune movement to address climate change, leading to 47% of Australians showing increased concern about climate change [3]. Furthermore, it was observed that this concern about climate change surpassed that of COVID-19 by three times [3]. Over time, the pandemic was perceived as an exacerbating factor of the climate crisis, creating an ongoing and evolving issue. Individual and collective behaviors during the pandemic exerted a powerful force that urged governments to protect the environment and respond effectively to the challenge of climate change [4]. Moreover, it is considered that during that time, the daily use of masks, gloves, and protective plastics made a negative impact on climate change because of the amount of generated waste.

Therefore, as allied healthcare professionals, podiatrists have a significant role in the community and can shape positive environmental practices, which can be effective in changing the behavior of a broader community, as seen in the last century when doctors stopped smoking [2]. As consumers of foot health products, it is increasingly likely that our patients expect more sustainable practices and products, including options for “green shoes”. In general, they are made with materials including recycled plastics and plant-based products. There is also a trend toward fairtrade and ethical production, and some companies pay carbon offsets for freight (and it is good to ‘buy local’ where possible) [2].

The term ‘green podiatry’ and the first publication about it was described by Angela Evans in 2021 [2]. Green podiatry, as part of sustainable healthcare, guides us to be responsible consumers of energy and products and to reduce emissions in the workplace. However, Spanish podiatrists are still unaware of these concepts and therefore do not apply them to improve the planet’s overall health [3].

As healthcare professionals, podiatrists have the ability to contribute to reducing climate change through actions such as prescribing exercise as the primary treatment option, as it is evidence-based and beneficial for all patients, instead of prescribing unnecessary orthotics. Additionally, choosing active transportation methods such as walking or cycling over using polluting modes of transportation can also make a significant impact [1].

Given the current lack of knowledge regarding this topic among Spanish podiatrists, the objective of this study was to assess the importance they place on climate change and whether they undertake actions to mitigate it and its repercussions.

## 2. Materials and Methods

### Participants

An online survey (via Google Forms) was designed to be completed by Spanish podiatrists. The inclusion criteria for the study were as follows: (1) Having a clinical practice in Spanish territory; (2) Regularly working in a podiatry clinic for at least 1 year; (3) Willing to participate voluntarily and providing explicit agreement (by clicking) in the informed consent to participate in the study. Participants would be excluded if they were podiatrists practicing outside of Spain.

The survey was sent through the professional podiatry associations in different autonomous communities of podiatrists that were able to answer in April 2023. The content and procedure of the survey were approved by the UEX Bioethics and Biosafety Committee (Registration No: 21/2023). The survey consisted of a series of questions including anthropometric data, Likert-scale questions, and short questions to assess attitudes in their work environment and their contribution to the process of climate change. 


1-to-5-point Likert-scale questions:
Do you know the concept of “Green Podiatry”?Do you worry about climate change?Do you know what the carbon footprint is?Do you think podiatrists have a major role in climate change?Do you think that podiatric attention could help contribute to improving climate change?Is there evidence that single-use items are always safer than reusable ones?Do you consider exercise as a first-line treatment?Have you prescribed any treatment without being necessary?Do you know about green shoes, made from natural fibers (raffia palm or banana skin)?Do you know about the recycling of shoes?Do you know the United Nations’ sustainable development goals?Do you think podiatrists are in a great position to act and educate people on climate change?


Short text questions:
13.If you know them, do you know in which of them the “green podiatry” could collaborate?14.How do you think we could act and educate on climate change?


Basic demographic information:
-Your age now?-How many years as a podiatrist? OR are you a podiatry student?-Do you own a car? Yes/No-If yes, is your car: electric/hybrid/fuel-Do you ride a bicycle? Yes/No-Do you use public transport (bus, train)? Yes/No-How many hours physical activity each week? (include walking, riding, gym, sports)-What is your BMI?-Are you vegetarian or vegan? Yes/No


Outcome Measures:


The following outcome measures will be assessed and analyzed:

-Gender distribution among participants.-Age range of participants.-Level of knowledge about green podiatry and sustainable footwear.-Commuting habits in relation to climate change.-Weekly physical activity level.-Frequency of prescribing unnecessary treatments to patients.-Use of physical activity as a first-line treatment option.-The responses will be tabulated to determine if specific practices are already being adopted by Spanish podiatrists.

## 3. Results

The survey was completed by 272 participants. The average age of the respondents was 39.4 ± 9.9 years, with an average working period in the clinic of 15.8 ± 9.3 years. The average height of the respondents was 1.68 ± 0.08 m, and the average weight was 68.2 ± 14.5 kg. Of the surveyed podiatrists, 91.5% owned a car, while only 8.5% reported not having one. Among those who owned a vehicle, 93.6% had a fossil-fuel-powered car, while only 0.4% had an electric one.

### 3.1. Transport and Physical Activity

Regarding commuting habits, 86% of the podiatrists surveyed did not use bicycles for transportation, while 14% did. Additionally, 30.9% of the respondents used public transportation, while 69.1% did not. In terms of physical activity, 30.9% dedicated 3–4 h per week to exercise, 23.9% dedicated 1–2 h, 20.6% dedicated 5–6 h, 9.6% dedicated more than 8 h, and 8.8% did not engage in physical activity (Figure 1). Regarding dietary habits, 4% of the respondents identified as vegans, while 96% did not follow a vegan diet. 

### 3.2. Degree of Knowledge in Podiatry and Sustainable Development Goals (SDGs)

A total of 89% of the respondents in this survey had never heard or come across the concept of “green podiatry”, and only 0.7% were familiar with it in-depth. However, nearly half of them, 47.8%, expressed significant concern about climate change. A total of 34.2% of the respondents were familiar with the concept and understand the consequences of the carbon footprint, while 19.5% had no knowledge about this topic (Figure 2). A total of 44.9% of the respondents remained neutral when asked if podiatrists play an important role in climate change, compared to 12.5% who believed they do have an important role and 11.8% who think they have no involvement. Similarly, 33.8% remained neutral when asked if podiatric care contributes to climate change (Figure 3).

### 3.3. Actions of Commitment to Climate Change

The majority of podiatrists who participated in this survey considered exercise as a first-line treatment. A total of 30.1% stated that they prescribe it to all their patients, while 2.2% said they never prescribe it. A total of 44.9% claimed to have never prescribed an unnecessary treatment, while 1.8% admitted to having done it frequently.

Among the surveyed podiatrists, 41.5% had no knowledge about “green shoes”, while 21.7% were familiar with these alternatives. A total of 48% of them had no knowledge about footwear recycling, while 1.5% claimed to be well-informed (Figure 4).

A total of 56.6% of the respondents had no knowledge about the United Nations’ goals, while 2.6% were well-acquainted with them.

The most common responses to the question of how podiatrists could act and educate people about climate change include:-Reducing single-use materials.-Educating ourselves about footwear recycling and sustainable footwear.-Promoting and prescribing physical exercise as a therapeutic measure.

## 4. Discussion

The Spanish podiatrists surveyed did not seem to have favorable travel habits to combat climate change, which contributes to a much faster advancement of exposing the planet to irreversible damages [5]. Opting for active means of transportation, such as walking or cycling, diminishes greenhouse gas emissions and enhances air quality. Moreover, the resultant physical activity confers health benefits, reducing the susceptibility to chronic diseases and lessening the strain on healthcare resources. Additionally, 62% of the Spanish podiatrists do not comply with the recommended guidelines for weekly physical activity established by the World Health Organization, which is 300 min per week [6]. Specifically, 32% engage in 0 to 120 min of physical activity, and 30.9% engage in 180 to 240 min. This can be attributed to the nature of their work, which is generally carried out in a single clinic, with minimal movement within clinic corridors. Furthermore, we must consider that the increase in motorization and demand for private vehicles may be reducing exercise, such as walking to work [7]. Similarly, the use of bicycles has a positive impact on the environment, helps reduce traffic congestion, and has a beneficial effect by promoting physical activity and reducing “ecological anxiety”, aligning with the Sustainable Development Goals (SDGs) [8].

Therefore, this is an adverse effect associated with the specific typology of podiatry work. In this regard, several studies have shown that physicians display a low level of physical activity. An illustrative example is a study conducted in Cameroon, where a significant 86% of physicians did not reach the recommended level of physical activity [9]. Additionally, another study conducted in Bahrain revealed that only 29.6% of physicians exercised for at least 30 min per week, and only 13% reported physical activity for a minimum of 5 days per week [10].

In the western region of Saudi Arabia, another study indicated that 65% of physicians were classified as physically inactive, with only 35% reporting engagement in any sports-related activity [11]. These findings underscore the importance of addressing the issue of physical activity within the medical community [12].

In comparison with our study, better physical activity behaviors have been observed in some studies involving younger healthcare professionals. For example, research conducted in the United States, Brazil, and Malaysia revealed that approximately 66%, 53.1%, and 55% of healthcare professionals, respectively, were physically active [13,14,15]. These studies also found that the average age of American physicians was 34 years, and 70% of healthcare professionals in Malaysia and Brazil were under 40 years old.

Lack of physical activity can have significant repercussions, such as cardiovascular diseases, as physical activity prevents artery hardening, obesity, respiratory problems, type 2 diabetes, and arthritis, among other conditions [16]. Recommendations could include taking breaks between patients to walk and walking or cycling to the clinic, among other measures.

On the other hand, for the first time at a global level, during the COP26 (2019 United Nations Climate Change Conference), podiatry was acknowledged as a transformative driving force. This endeavor is spearheaded by the Australian Podiatry Association, with a focus on sustainability. Diverse perspectives from industry, science, medicine, sports, fashion, and retail are being integrated, underscoring the importance of active collaboration to promote a more sustainable lifestyle [1].

The concept of “Green Podiatry” stands out as it aligns with other medical groups in the pursuit of ecologically conscious solutions. In this context, the United Kingdom’s NHS-affiliated podiatrists have served as a source of inspiration to foster more sustainable practices within the podiatric profession [1]. Podiatrists in Spain have not yet fully grasped the scope and potential. There is a conspicuous lack of complete commitment to climate change. The survey data demonstrate an overall unfamiliarity with the term “green podiatry”, coupled with a failure to engage in selective waste recycling within their clinics. Nevertheless, the respondents exhibit a dedication to the future, as they express concern regarding climate change and its ramifications. In certain hospitals, initiatives are already being deployed, such as waste recycling, the sustainable management of hospital resources, and the incorporation of renewable energy sources for heating and water pumping [17].

Greater leadership is required, both from the General Council of Podiatry Colleges of Spain (CGCPE) and from governmental institutions, to drive a shift in these habits. This would enable a reduction in excessive consumption and the application of circular economy principles. Purchasing decisions, including footwear acquisition, could be transformed into a more environmentally respectful cycle.

Climate-conscious podiatrists raise awareness within the community and promote habits akin to those practiced within hospitals. In this vein, a study focused on pediatric flatfoot indicates that no disparity exists between children who have undergone treatment for arch development and those who have not. Similarly, the use of orthopedic shoes often proves uncomfortable for the child and does not yield superior development compared to the natural progression. Consequently, this translates into a reduced consumption of unnecessary and costly materials from an environmental perspective. Thus, the emphasis is placed on enhanced resource utilization, coupled with the implementation of more streamlined processes [18]. This approach advocates for the utilization of exceptionally versatile materials in orthotic fabrication [19], complemented by the integration of renewable raw materials containing components derived from vegetable oils, thereby mitigating polyol production in polyurethane development [20].

Regarding the fact that the majority of surveyed podiatrists prioritize exercise as the primary treatment, no changes would be recommended, as they are acting correctly in this regard. It has been demonstrated that physicians who engage in physical activity and dedicate at least 3 min per visit to providing exercise advice achieved a 25% adherence rate among their patients to these recommendations [12]. This finding reinforces the notion that physicians who do not adopt healthy behaviors are less likely to advise their patients on healthy lifestyle habits. In fact, a study revealed that physically inactive physicians are less likely to offer recommendations on physical activity to their patients [21].

Actions that could be implemented to be more sustainable in our clinic include:-Reducing single-use materials, as a significant amount of plastic waste is generated.-Not immediately prescribing foot orthotics without first trying specific exercises for the pathology.-Organizing workshops or lectures for podiatrists on footwear recycling and “green footwear” to raise awareness and be able to recommend them to patients.

Each one of us shares the responsibility to work as sustainably as possible and make our best efforts to live “green”. This includes the actions we take as podiatrists (reducing waste, minimizing unnecessary treatments, promoting and practicing “green podiatry”) and as citizens (buying local products, consuming less, choosing renewable energy sources, reducing the use of fossil fuels, managing waste). It will be very important to have the opportunity to repeat this (or a similar survey) in other places, for example, in the United Kingdom and Australia, and use it as a “baseline” to measure the effects of education in “green podiatry”.

This study represents only a preliminary analysis and overview of the data collected. The research team will prepare a report with the results of the study and later will transmit it both to the General Council of Colleges of Podiatrists in Spain and to the network of regional podiatry colleges. The purpose of this effort is to facilitate the dissemination of our recommendations among its affiliates. This will promote a positive impact on climate change, even if it is small, immediately in the actions carried out by podiatrists.

## 5. Conclusions

-The podiatrists who responded to the survey do not engage in sustainable commuting practices.-Spanish podiatrists are unaware of the concept of “green podiatry”, although they express significant concern about climate change. However, they seem to have some knowledge about carbon footprints. Almost half of them are unsure if podiatry plays an important role in climate change.-The actions of the surveyed podiatrists, such as prioritizing exercise as a primary treatment and avoiding unnecessary treatments, are favorable for addressing climate change. However, the majority of them have no knowledge about footwear recycling and “green shoes”. Spanish podiatrists prioritize reducing single-use materials and express interest in learning about footwear recycling and sustainable shoes.

Two recommendations for podiatrists could be:Embrace the three pillars of green podiatry: exercise, evidence, and daily practice.Utilize walking as a carbon-neutral means of transportation and promote physical activity as a means to improve health.

All Spanish podiatrists must channel their efforts towards reducing and altering their consumption habits to mitigate their carbon footprint.

## Figures and Tables

**Figure 1 ijerph-20-06761-f001:**
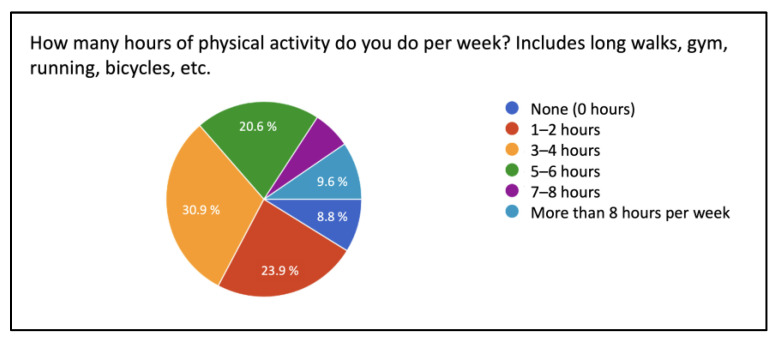
Weekly physical activity.

**Figure 2 ijerph-20-06761-f002:**
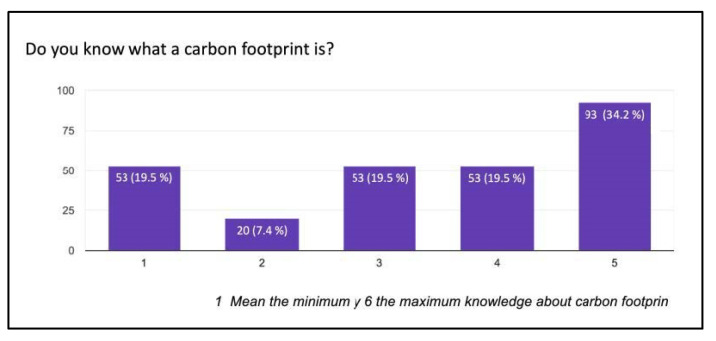
Knowledge about the carbon footprint.

**Figure 3 ijerph-20-06761-f003:**
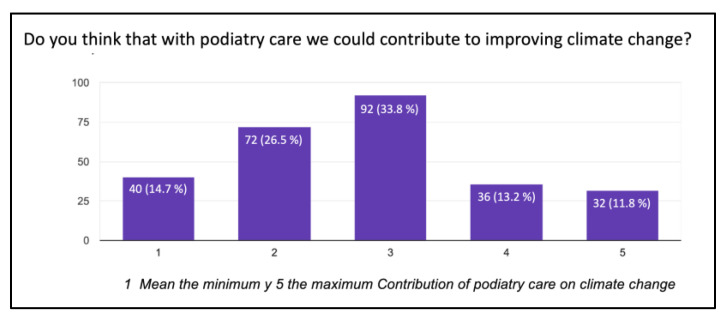
Contribution of podiatry care on climate change.

**Figure 4 ijerph-20-06761-f004:**
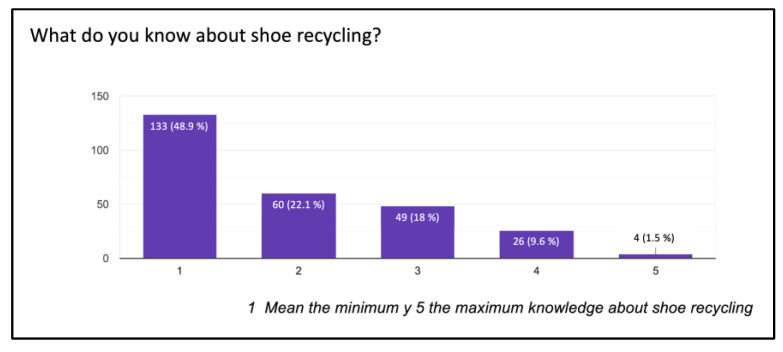
Shoe recycling.

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
