# Peer review of "Degree of Knowledge and Commitment of the Spanish Podiatry Community to Green Podiatry: A Preliminary Report"

_ijerph, 2023, doi:10.3390/ijerph20186761_

Round 1
Reviewer 1 Report
Dear authors, thank you for the opportunity to review this article. Nowadays, it is essential to take into consideration the topic of climate change. Therefore, to aim your investigation into assessing the level of awareness among podiatrists regarding this issue and determine their attitudes toward the climate change process is vital.
Line 6: The PubMed/MEDLINE standard format used for affiliations: complete address information including city, zip code, state/province, and country.
Line 45: Could you provide a wider explanation about the relationship between the COVID-19 pandemic and its positive role in the climate change? It is considered that during that time, the daily use of masks, gloves and protective plastics made a negative impact on the climate change because of the amount of generated waste.
Line 54: Could you define the concept of “green shoes”?
Line 80: Which previous studies inspired you to develop the selected questions? Or your own experience?
Line 119: I consider that the bullet point should be removed to avoid confusion.
Figure 2: Does 6 mean the maximum knowledge about carbon footprint and 1 the minimum knowledge? That should be clarify under the figure. The same with the figure 3, etc.
Line 197: What is the relationship between the weekly physical activity (i.e. going to the gym or swimming) and the climate change? Maybe you tied to stablish that relationship with the sentence contains between the lines 255 and 258.
Author Response
Dear Reviewer´s
Thank you for your kind and very helpful comments. I have looked through the manuscript following your recommendations and suggestions.
I will show to you the changes we have made due to your review, which are underlined in yellow in the paper.
Reviewer #1:
Dear authors, thank you for the opportunity to review this article. Nowadays, it is essential to take into consideration the topic of climate change. Therefore, to aim your investigation into assessing the level of awareness among podiatrists regarding this issue and determine their attitudes toward the climate change process is vital.
Line 6: The PubMed/MEDLINE standard format used for affiliations: complete address information including city, zip code, state/province, and country.
Response: we have completed this info
Line 45: Could you provide a wider explanation about the relationship between the COVID-19 pandemic and its positive role in the climate change? It is considered that during that time, the daily use of masks, gloves and protective plastics made a negative impact on the climate change because of the amount of generated waste.
Response: we have explained and expanded the relationship between the COVID-19 pandemic and its positive
Line 54: Could you define the concept of “green shoes”?
Response: we have included definition the concept of “green shoes in the text
Line 80: Which previous studies inspired you to develop the selected questions? Or your own experience?
Response: The design of our study has been conceived by ourselves, drawing upon our own experience and independent of prior studies.
Line 119: I consider that the bullet point should be removed to avoid confusion.
Response: we have proceded to eraser the bullet
Figure 2: Does 6 mean the maximum knowledge about carbon footprint and 1 the minimum knowledge? That should be clarify under the figure. The same with the figure 3, etc.
Response: we have clarified the figures
Line 197: What is the relationship between the weekly physical activity (i.e. going to the gym or swimming) and the climate change? Maybe you tied to stablish that relationship with the sentence contains between the lines 255 and 258.
Response: we have expanded this relationship in the text
Reviewer 2 Report
Dear authors,
Thank you for allowing me to review your paper on this timely and important subject.
I think that the survey used for your study collects very interesting data and asks pertinent questions. However, I also believe that the analysis and the presentation of the data can tremendously improve. You collected disaggregated data, which contains an enormous amount of information, but do not fully benefit from this knowledge. I would recommend looking at differences between age groups and gender in terms of usage of public transport, daily exercise levels and BMI and weight, instead of just offering a descriptive reporting of the data.
The discussion section focuses a lot on the benefits of daily exercise and outlines the health benefits of doing so. The discussion section as such, seems to divert from the initial intention of the paper and does not offer a true discussion of the knowledge and commitment of podiatrists to green podiatry in Spain.
Additionally, even though individual efforts towards mitigating climate change are important, they will not do anything without actual binding commitments from governments and large corporations. I am missing a larger systems-based approach to the value of green podiatry and how podiatry as a field can contribute to sustainable development and clean energy.
Last but not least, I am missing a true tangible message being delivered to the podiatry community or the larger health field in general.
Author Response
Dear Reviewer´s
Thank you for your kind and very helpful comments. I have looked through the manuscript following your recommendations and suggestions.
I will show to you the changes we have made due to your review, which are underlined in blue in the paper.
Reviewer #2:
Dear authors,
Thank you for allowing me to review your paper on this timely and important subject.
Response: thank you for your dedication.
- I think that the survey used for your study collects very interesting data and asks pertinent questions. However, I also believe that the analysis and the presentation of the data can tremendously improve. You collected disaggregated data, which contains an enormous amount of information, but do not fully benefit from this knowledge. I would recommend looking at differences between age groups and gender in terms of usage of public transport, daily exercise levels and BMI and weight, instead of just offering a descriptive reporting of the data.
Response: we have presented this work as a preliminary study. All of these data are being rigorously analyzed to be presented in a subsequent study with greater detail and scope.
- The discussion section focuses a lot on the benefits of daily exercise and outlines the health benefits of doing so. The discussion section as such, seems to divert from the initial intention of the paper and does not offer a true discussion of the knowledge and commitment of podiatrists to green podiatry in Spain.
Response: we have expanded the discussion section with a greater commitment of podiatrists to ecology
- Additionally, even though individual efforts towards mitigating climate change are important, they will not do anything without actual binding commitments from governments and large corporations. I am missing a larger systems-based approach to the value of green podiatry and how podiatry as a field can contribute to sustainable development and clean energy.
Response we have enhanced the discourse pertaining to the contribution of sustainable development and clean energy within the discussion section.
- Last but not least, I am missing a true tangible message being delivered to the podiatry community or the larger health field in general.
Response we have added a direct message for the podiatrist community in conclusion section
Round 2
Reviewer 2 Report
Dear authors,
Thank you for sharing your updated manuscript with and for taking the reviewer's feedback into consideration.
I think the manuscript has improved a lot. I just have a few remaining suggestions below:
- I would reassess if all concept discussed in the discussion section have been sufficiently introduced in the introduction. I'm thinking about UNFCCC COP meetings among other topics that may require some additional background for IJERPH's readership to fully grasp the impact.
- I would add a paragraph outlining the next steps that you and your team plan to undertake with this data and explaining that this analysis is just a preliminary analysis and baseline overview of the data collected.
Author Response
Dear Reviewer´s
Thank you for your kind and very helpful comments. I have looked through the manuscript following your recommendations and suggestions.
I will show to you the changes we have made due to your review, which are underlined in blue in the paper.
Reviewer #2:
Comments and Suggestions for Authors
Dear authors,
Thank you for sharing your updated manuscript with and for taking the reviewer's feedback into consideration.
I think the manuscript has improved a lot. I just have a few remaining suggestions below:
1.- I would reassess if all concept discussed in the discussion section have been sufficiently introduced in the introduction. I'm thinking about UNFCCC COP meetings among other topics that may require some additional background for IJERPH's readership to fully grasp the impact
Response: we have added more information for a better understanding of IJERPH readers in the introduction section
2.- I would add a paragraph outlining the next steps that you and your team plan to undertake with this data and explaining that this analysis is just a preliminary analysis and baseline overview of the data collected.
Response: we have incorporated your suggestion about the steps to take and that it is a preliminary study in the text